# Conductance-based dendrites perform Bayes-optimal cue integration

**Jakob Jordan** [1,2]*, **João Sacramento**[1,3], **Willem A. M. Wybo** [1,4], **Mihai A. Petrovici**[1‡], **Walter Senn**[1‡]

**1** Department of Physiology, University of Bern, Bern, Switzerland, **2** Electrical Engineering, Yale University, New Haven, Connecticut, United States of America, **3** Institute of Neuroinformatics, UZH / ETH Zurich, Zurich, Switzerland, **4** Institute of Neuroscience and Medicine, Forschungszentrum Jülich, Jülich, Germany

‡ These authors are joint senior authors on this work.
* jakob.jordan@unibe.ch

**Data Availability Statement:** Data and code is available from https://github.com/unibe-cns/learning-bayes-optimal-dendritic-opinion-pooling.

**Funding:** This work has received funding from the European Union 7th Framework Programme under

## Abstract

A fundamental function of cortical circuits is the integration of information from different sources to form a reliable basis for behavior. While animals behave as if they optimally integrate information according to Bayesian probability theory, the implementation of the required computations in the biological substrate remains unclear. We propose a novel, Bayesian view on the dynamics of conductance-based neurons and synapses which suggests that they are naturally equipped to optimally perform information integration. In our approach apical dendrites represent prior expectations over somatic potentials, while basal dendrites represent likelihoods of somatic potentials. These are parametrized by local quantities, the effective reversal potentials and membrane conductances. We formally demonstrate that under these assumptions the somatic compartment naturally computes the corresponding posterior. We derive a gradient-based plasticity rule, allowing neurons to learn desired target distributions and weight synaptic inputs by their relative reliabilities. Our theory explains various experimental findings on the system and single-cell level related to multi-sensory integration, which we illustrate with simulations. Furthermore, we make experimentally testable predictions on Bayesian dendritic integration and synaptic plasticity.

## Author summary

The only certainty is uncertainty. Whether it is the reconstruction of a three-dimension scene from the two-dimensional images on our retina or locating your lock in twilight, we have to make decisions and perform actions without knowing the exact state of our environment. In the presence of uncertainty, Bayesian probability theory provides formal recipes of how different pieces of information should be combined to gain maximal information. Indeed, behavioral experiments show that humans and other animals behave as if they operate according to these principles. However, so far it is unclear how the necessary computations are implemented by our biological substrate. By suggesting a new view on the dynamics of a broad class of neuron models, we show how these computations may be implemented by individual cortical neurons. Furthermore, we derive a novel

grant agreement 604102 (HBP), the Horizon 2020 Framework Programme under grant agreements 720270, 785907 and 945539 (HBP), the Swiss National Science Foundation (SNSF, Sinergia grant CRSII5-180316) and the Manfred Stärk Foundation. The funders had no role in study design, data collection and analysis, decision to publish, or preparation of the manuscript.

**Competing interests:** The authors have declared that no competing interests exist.

model of synaptic plasticity from first principles and illustrate how a neuron equipped with these synapse dynamics learns to approximate Bayes-optimal decision makers. Finally, we interpret various experimental results in light of our proposed theory and make experimentally testable predictions.

## Introduction

Successful actions are based on information gathered from a variety of sources. This holds as true for individuals as it does for whole societies. For instance, experts, political parties, and special interest groups may all have different opinions on proposed legislature. How should one combine these different views? One might, for example, weight them according to their relative reliability, estimated from demonstrated expertise. According to Bayesian probability theory, the combined reliability-weighted view contains more information than any of the individual views taken on its own and thus provides an improved basis for subsequent actions [1].

Such problems of weighting and combining information from different sources are commonplace for our brains. Whether inputs from neurons with different receptive fields or inputs from different modalities (Fig 1a), our cortex needs to combine these uncertain information sources into a coherent basis that enables informed actions.

Bayesian probability theory provides clear recipes for how to optimally solve such problems, but so far the implementation in the biological substrate is unclear. Previous work has demonstrated that multiple interacting neuronal populations can efficiently perform such probabilistic computations [3, 4]. These studies provided mechanistic models potentially underlying the often Bayes-optimal behavior observed in humans and other animals [2, 5, 6]. Here we demonstrate that probabilistic computations may be even deeper ingrained in our biological substrate, in single cortical neurons.

We suggest that each dendritic compartment, here interpreted as logical subdivision of a complex morphology, represents either a (Gaussian) likelihood function or a (Gaussian) prior distribution over somatic potentials. These are parametrized by the local effective reversal potential and the membrane conductance. Basal dendrites receiving bottom-up input represent likelihoods, while apical dendrites receiving top-down input, represent priors. We show that the natural dynamics of leaky integrator models compute the corresponding posterior. The crucial ingredient is the divisive normalization of compartmental membrane potentials

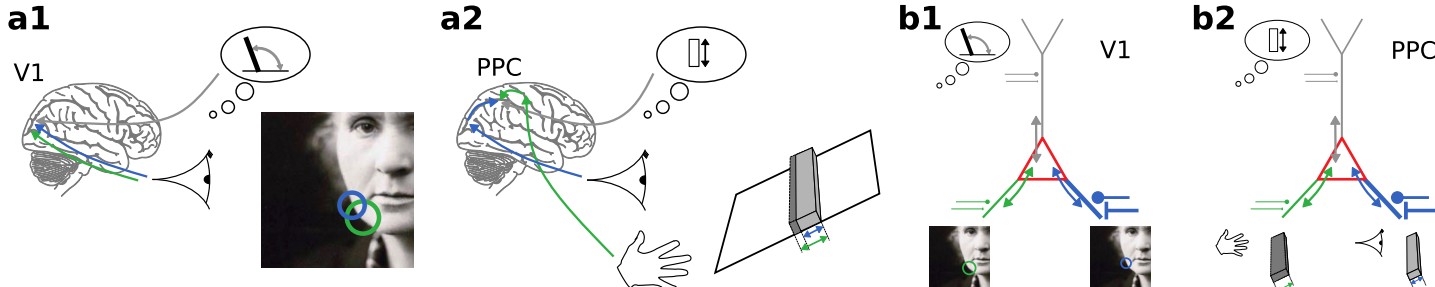

**Fig 1. Integration of uncertain information in cortical neurons. (a1)** Cue integration in early visual processing judging the orientation of a local edge. **(a2)** Cue integration in multimodal perception judging the height of a bar [2]. **(b1)** A neuron integrates visual cues and prior expectations to combine information across receptive fields. **(b2)** A neuron integrates visual and haptic cues with prior expectations to combine information across modalities. These computations can be realized by the natural dynamics of cortical neurons through the bidirectional coupling of compartments (colored arrows) which represent likelihood functions (green, blue), prior (grey), or posterior distributions (red) through their local membrane conductance and effective reversal potential.

naturally performed in the presence of conductance-based synaptic coupling [7]. Furthermore, while this computation relies on bidirectional coupling between neuronal compartments, at the level of the neuronal input-output transfer function, the effective computation can be described in a feed-forward manner.

Beyond performing inference, the single-neuron view of reliability-weighted integration provides an efficient basis for learning. In our approach, synapses not only learn to reproduce a somatic target activity [8], but they also adjust synaptic weights to achieve some target variance in the somatic potential. Furthermore, afferents with low reliability will be adjusted to contribute with a smaller total excitatory and inhibitory conductance to allow other projections to gain more influence. Implicitly, this allows each dendritic compartment to adjust its relative reliability according to its past success in contributing to matching desired somatic distributions.

In our theoretical framework we derive somatic membrane potential dynamics and synaptic plasticity jointly via stochastic gradient ascent on the log-posterior distribution of somatic potentials. Simulations demonstrate successful learning of a prototypical multisensory integration task. The trained model allows us to interpret behavioral and neuronal data from cue integration experiments through a Bayesian lens and to make specific predictions about both system behavior and single cell dynamics.

## Results

### Integration of uncertain information in cortical neurons

To give a high-level intuition for our approach, let us consider a prototypical task our brains have to solve: the integration of various cues about a stimulus, for example in early visual areas from different parts of the visual field (Fig 1a) or in association areas from different sensory modalities (Fig 1b). Due to properties of the stimulus and of our sensory systems, information delivered via various modalities inherently differs in reliability. Behavioral evidence demonstrates that humans and non-human animals are able to integrate sensory input from different modalities [2, 5, 6, 9–14] and prior experience (e.g., [15, 16]), to achieve a similar performance as Bayes-optimal cue-integration models. Our theory suggests that pyramidal cells are naturally suited to implement the necessary computations. In particular they take both their inputs and their respective reliabilities into account by using two orthogonal information channels: membrane potentials and conductances.

Consider a situation where your visual sensory apparatus is impaired, for example, due to a deformation of the lens. Presented with multimodal stimuli that provide auditory and visual cues, you would have learned to rely more on auditory cues rather than visual input (Fig 2). When confronted with an animal as in Fig 2a, based on your vision alone, you might expect it to be a cat, but not be certain about it. Hearing it bark, however, would shift your belief towards it being, with high certainty, a dog. Since current-based neuron models only encode information about their preferred feature in the total synaptic current without considering the relative reliability of different pathways, they can generate wrong decisions: here, a neuron that integrates auditory and visual cues wrongly signals the presence of a cat to higher cortical areas (Fig 2b). In contrast, as we will show in the next section, by using dendritic conductances $g^d$ as an additional coding dimension besides effective dendritic reversal potentials $E^d$, conductance-based neuron models are able to respond correctly by weighting auditory inputs stronger than visual inputs (Fig 2c). Intuitively, in the absence of stimuli, the "cat neuron" (Fig 2b and 2c) represents a small (prior) probability that a cat may be present, and the presentation of an ambiguous cat-dog image increases this probability (Fig 2e, 400–1200ms, d,e). However, when the animal subsequently barks, the probability drops abruptly. In our approach these

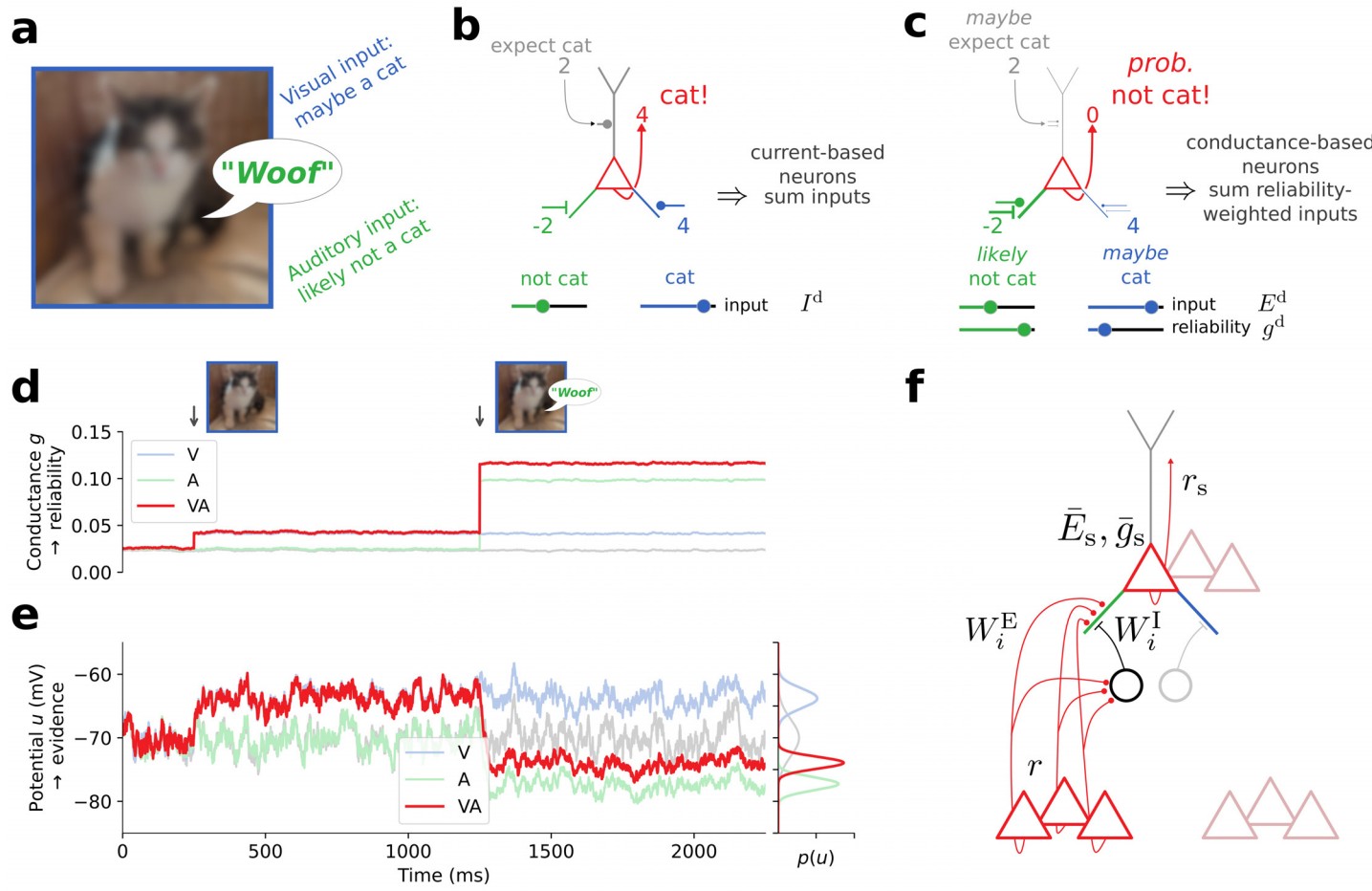

**Fig 2. Conductance-based neuronal dynamics naturally implement Bayesian cue integration. (a)** A multisensory stimulus. **(b)** Current-based neuron models can only additively accumulate information about their preferred feature. **(c)** Conductance-based neuron models simultaneously represent information and associated reliability. **(d)** Total somatic conductances $\bar{g}_s$ consisting of leak and synaptic conductances in a multisensory neuron (see panel (c)) under three conditions: only visual input (V, blue), only auditory input (A, green), bimodal input (VA, red), and no input (gray). Before 400ms the visual cue is absent. Before 1200ms the auditory cue is absent. **(e)** Somatic membrane potentials $u_s$ are noisy, time-continuous processes that sample from the somatic distributions in the respective condition. This histogram on the right shows the somatic potential distributions between 1250ms and 2250ms. **(f)** Suggested microcircuit implementation. Top part shows the neuron from panel (c). Activity $r$ of pyramidal cells from lower areas is projected directly (red lines with circular markers, $W_i^E$ denote excitatory synaptic weights) and indirectly via inhibitory interneurons (circles and black lines with bar markers, $W_i^I$ denote inhibitory synaptic weights) to different dendritic compartments of pyramidal cells in higher cortical areas. Each pyramidal cell represents pooled information $\bar{E}_s$ with its associated reliability $\bar{g}_s$ distributed across a corresponding population (overlapping triangle triples, representing pre- and postsynaptic neurons, respectively).

computations are reflected by a hyperpolarization of the somatic membrane potential and an associated increase in membrane conductance. Consistent with Bayes-optimal cue-integration models (e.g., [17]), the combined estimate shows an increased reliability, even if the cues are opposing.

## Bayesian neuronal dynamics

Excitatory and inhibitory conductances targeting a single microscopic neuronal compartment (with at most one excitatory and one inhibitory afferent) combine with the leak and the associated reversal potentials into a total transmembrane current $I^d = g^d(E^d - u^d)$. This current

induces a stimulus-dependent effective reversal potential $E^{\mathrm{d}}$ given by

$$E^{\mathrm{d}} = \frac{g^{\mathrm{E}}E^{\mathrm{E}} + g^{\mathrm{I}}E^{\mathrm{I}} + g^{\mathrm{L}}E^{\mathrm{L}}}{g^{\mathrm{E}} + g^{\mathrm{I}} + g^{\mathrm{L}}} \ , \tag{1}$$

where excitatory, inhibitory and leak reversal potential are denoted as $E^{\mathrm{E/I/L}}$, and the respective conductances by $g^{E/I/L}$. The sum of these three conductances $g^{\mathrm{d}} = g^{\mathrm{E}} + g^{\mathrm{I}} + g^{\mathrm{L}}$ represents the local membrane conductance, which excludes the coupling to other compartments. The excitatory and inhibitory conductances are the product of the synaptic weights times the presynaptic firing rates, $g^{\mathrm{E/I}} = W^{\mathrm{E/I}}r$. Note that in general $E^{\mathrm{d}}$ is different from the actual dendritic potential $u^{\mathrm{d}}$, which is additionally influenced by the membrane potential in neighboring compartments.

Across the dendritic tree (with multiple compartments $i$) we now interpret $g_i^{\mathrm{d}}$ and $E_i^{\mathrm{d}}$ as parameters of Gaussian [18] likelihood functions $p(E_i^{\mathrm{d}}|u_{\mathrm{s}}, g_i^{\mathrm{d}})$ in basal compartments and parameters of Gaussian priors $p(u_{\mathrm{s}}|E_i^{\mathrm{d}}, g_i^{\mathrm{d}})$ in apical compartments. The dendritic likelihoods quantify the statistical relationship between dendritic and somatic potentials. Intuitively speaking, they describe how compatible a certain somatic potential $u_{\mathrm{s}}$ is with an effective reversal potential $E_i^{\mathrm{d}}$. Note that this relation is of purely statistical, not causal nature—biophysically, effective reversal potentials $E_i^{\mathrm{d}}$ cause somatic potentials, not the other way around.

Finally, the somatic compartment computes the posterior according to Bayes theorem (see Methods Sec. "Bayesian theory of somatic potential dynamics" for details),

$$p(u_{\mathrm{s}}|\boldsymbol{W}, \boldsymbol{r}) \propto \text{likelihood} \times \text{prior} = e^{-\frac{\bar{g}_{\mathrm{s}}}{2\lambda_{\mathrm{e}}}(u_{\mathrm{s}} - \bar{E}_{\mathrm{s}})^2} \ . \tag{2}$$

Here, $\bar{g}_{\mathrm{s}}$ represents the total somatic conductance, and $\bar{E}_{\mathrm{s}}$ the total somatic reversal potential, which is given by the convex combination of the somatic and dendritic effective reversal potentials, weighted by their respective membrane conductances and dendro-somatic coupling factors (Fig 3). The "exploration parameter" $\lambda_{\mathrm{e}}$ relates conductances to membrane potential fluctuations. In general, this parameter depends on neuronal properties, for example, on the amplitude of background inputs and the spatial structure of the cell. It can be determined experimentally by an appropriate measurement of membrane potentials from which both fluctuation amplitudes and decay time constants $\tau = C/\bar{g}_{\mathrm{s}}$ can be estimated.

To obtain the somatic membrane potential dynamics, we propose that the soma performs noisy gradient ascent on the log-posterior,

$$\begin{aligned}
C\dot{u}_{\mathrm{s}} &= \lambda_{\mathrm{e}} \frac{\partial}{\partial u_{\mathrm{s}}} \log p(u_{\mathrm{s}}|\boldsymbol{W}, \boldsymbol{r}) + \xi \\
&= \bar{g}_{\mathrm{s}} (\bar{E}_{\mathrm{s}} - u_{\mathrm{s}}) + \xi \\
&= g_0(E_0 - u_{\mathrm{s}}) + \sum_{i=1}^{D} \alpha_i^{\mathrm{sd}} [g_i^{\mathrm{L}}(E^{\mathrm{L}} - u_{\mathrm{s}}) + g_i^{\mathrm{E}}(E^{\mathrm{E}} - u_{\mathrm{s}}) + g_i^{I}(E^{\mathrm{I}} - u_{\mathrm{s}})] + \xi \ .
\end{aligned} \tag{3}$$

with membrane capacitance $C$, and dendro-somatic coupling factors $\alpha_i^{\mathrm{sd}} = g_i^{\mathrm{sd}}/(g_i^{\mathrm{sd}} + g_i^{\mathrm{d}})$ that result from the dendro-somatic coupling conductances $g_i^{\mathrm{sd}}$ and the isolated dendritic conductances $g_i^{\mathrm{d}}$. The additive noise $\xi$ represents white noise with variance $2C\lambda_{\mathrm{e}}$, arising, for example, from unspecific background inputs [19–22]. For fixed presynaptic activity $r$, the average somatic membrane potential hence represents a maximum-a-posteriori estimate (MAP, [17]), while its variance is inversely proportional to the total somatic conductance $\bar{g}_{\mathrm{s}}$. The effective time constant of the somatic dynamics is $\tau = C/\bar{g}_{\mathrm{s}}$, thus enabling $u_{\mathrm{s}}$ to converge faster to reliable MAP estimates for larger $\bar{g}_{\mathrm{s}}$.

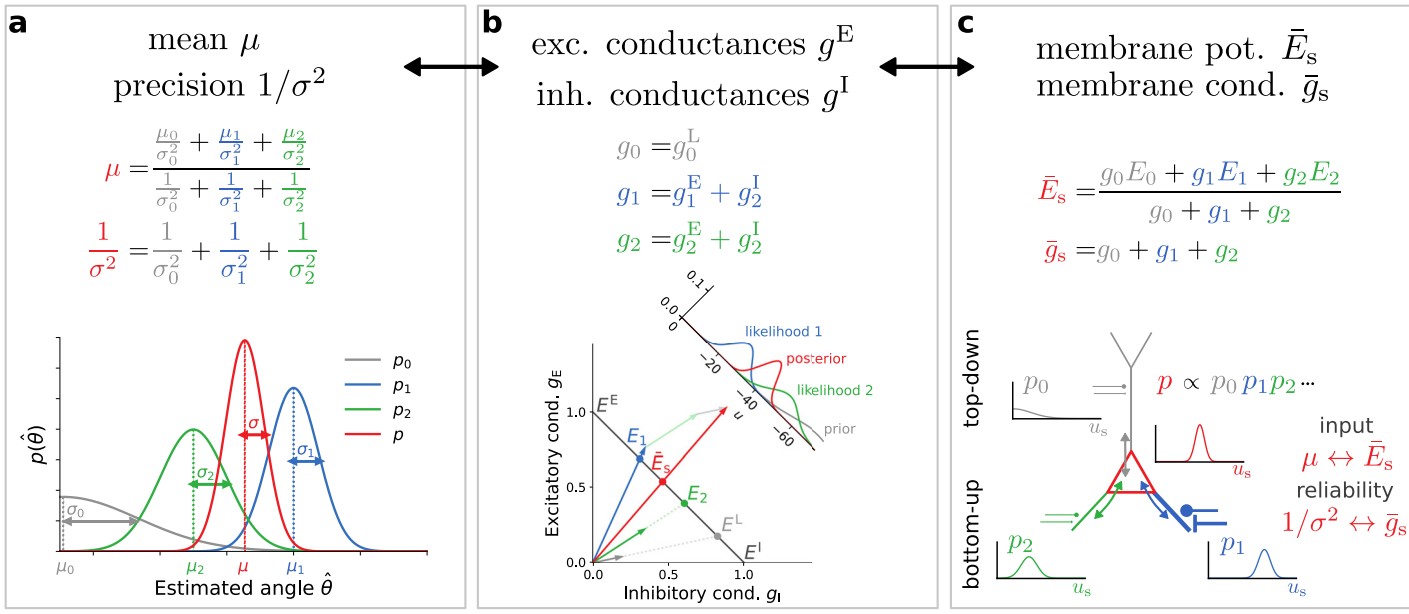

**Fig 3. Non-linear cue integration is achieved through a linear vector summation of conductances.** (a) Non-linear combination of Gaussian probability densities. The pooled mean is a convex combination of the original means, while the pooled reliability, the inverse variance, is a sum of the individual reliabilities. (b) Stimulus-evoked excitatory and inhibitory synaptic conductances as two-dimensional vectors (blue and green), as well as the leak (gray), are linearly summed across dendrites to yield the total somatic conductances (red arrow). The intersections with the antidiagonal (black line) yield the corresponding dendritic and somatic reversal potentials. This intersection is a nonlinear operation (see Methods Sec. "Linear coordinates for nonlinear processing"). The inset shows the full distributions. Note that the prior can be modulated by synaptic conductance elicited by top-down input (see panel c). (c) Translation of prior (gray) and dendritic (green and blue) potentials and conductances into the corresponding somatic mean potential and conductances (red). For visualization purposes, the prior distribution is only partially shown.

The dynamics derived here from Bayesian inference (Eq 3) are identical to the somatic membrane potential dynamics in bidirectionally coupled multi-compartment models with leaky integrator dynamics and conductance-based synaptic coupling (Fig 4) under the assumption of fast dendritic responses [23]. In other words, the biophysical system computes the posterior distribution via its natural evolution over time. This suggests a fundamental role of conductance-based dynamics for Bayesian neuronal computation.

Conductance-based Bayesian integration, as introduced above, can also be viewed from a different perspective in terms of probabilistic opinion pooling [24]. Under this view each

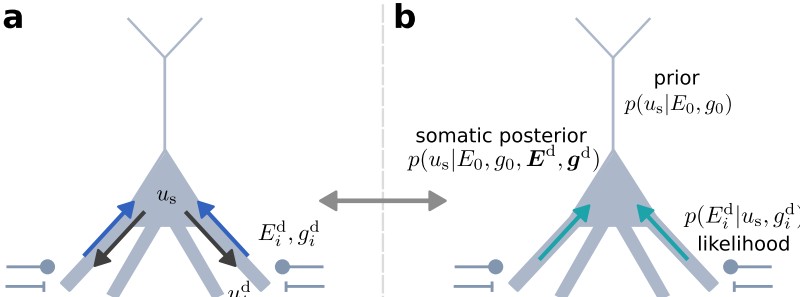

**Fig 4. Single neuron dynamics as Bayesian inference.** (a) Somatic and dendritic membrane potentials are coupled through currents flowing along the dendritic tree (blue and black arrows, Eqs 5 and 6). (b) The steady state of the somatic compartment can be interpreted as computing the posterior $p(u_s|E_0, g_0, \boldsymbol{E}^d, \boldsymbol{g}^d)$ from the dendritic priors $p(u_s|E_0, g_0)$ and dendritic likelihoods $p(E_i^d|u_s, g_i^d)$. Stimulus-driven effective reversal potentials in basal dendrites pull the somatic potential distribution from the prior towards the posterior.

dendrite can be thought of as an individual with a specific opinion—the dendrite's effective reversal potential—along with an associated reliability—the dendrite's conductance. Accordingly, the soma then plays the role of a "decision maker" that pools the reliability-weighted dendrite's opinions, determines a compromise, and communicates this outcome to other individuals, i.e., downstream neurons' dendrites. Intuitively speaking, in this process dendrites with a lot of confidence in their opinion, i.e., those with high dendritic conductance, contribute more to the pooled opinion than others.

Before introducing synaptic plasticity, we first discuss a specific consequence for neuronal dynamics arising from our Bayesian view of neuronal dynamics.

### Stimuli lead to Bayesian updates of somatic membrane potential statistics

The conductance-based Bayesian integration view predicts neuronal response properties that differ from those of classical neuron models. In the case of conductances, somatic membrane potentials reflect prior expectations in the absence of sensory input. These priors typically have low reliability, encoded in relatively small conductances. As a consequence, the neuron is more susceptible to background noise, resulting in large membrane potential fluctuations. Upon stimulus onset, presynaptic activity increases causing synaptic conductances to increase, thereby pulling postsynaptic membrane potentials towards the cue-specific reversal potentials $E^d$, irrespective of their prior value (Fig 5a). This phenomenon is observed in electrophysiological recordings from mouse somatosensory cortex: the change in membrane potential upon whisker stimulation pulls the somatic membrane potential from variable pre-stimulus potentials, i.e., different prior expectations, towards a cue-specific post-stimulus potential (Fig 5a, [25]). Besides a change in the average membrane potential, cue onset increases conductances and hence decreases membrane potential variability.

These effects are signatures of Bayesian computations. Upon cue onset, the prior distribution is combined with stimulus-specific likelihoods leading to an updated somatic distribution with adapted mean and reduced variance. If the prior strongly disagrees with information provided by the stimulus, the change in mean is larger than if prior and stimulus information are consistent. Importantly, the variance is always reduced in the presence of new information, regardless of whether it conflicts with previous information or not; this is a hallmark of Bayesian reasoning.

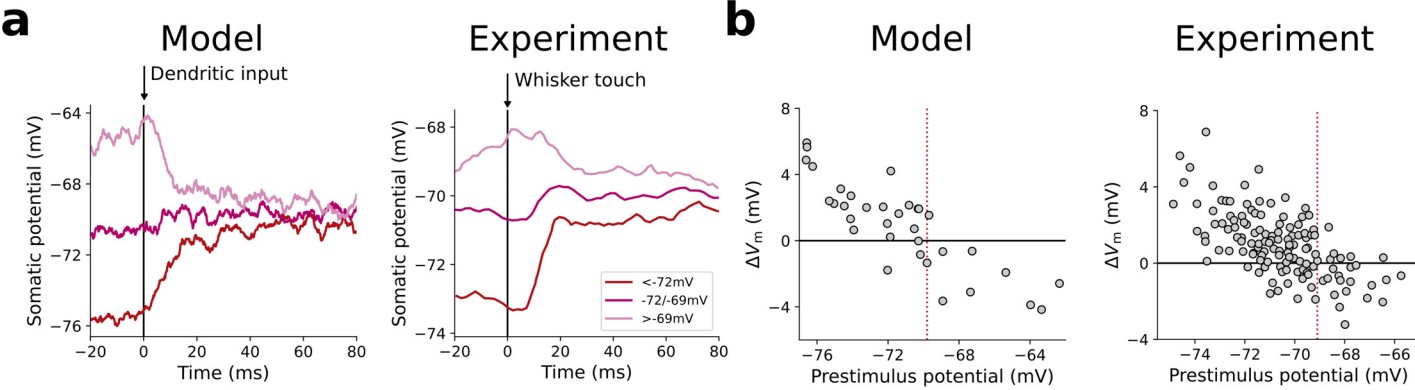

**Fig 5. Conductance-based Bayesian integration implies stimulus-specific reversal potentials. (a)** Average stimulus-evoked responses for different ranges of prestimulus potentials generated by our model (left) and measured experimentally (right, see [25]). Vertical arrow indicates stimulus onset corresponding to activation of dendritic input and whisker touch, respectively. Independently of the previous value of the somatic potential, the dendritic input always pulls the somatic potential towards the effective reversal potential associated with the stimulus. **(b)** PSP amplitude vs. prestimulus potential generated by our model (left) and measured experimentally (right, see [25]). Experiment data from [25].

We propose that this probabilistic computation underlies the observed stimulus-driven reduction of variability throughout cortex [26, 27] and explains why stimulus-evoked PSP amplitudes are negatively correlated with prestimulus potentials [25, 28, Fig 5b; also see]. In whisker stimulation experiments [25], the stimulation intensity is encoded by the whisker deflection angle. Our framework predicts that, as the amplitude of whisker deflections increases, the variance of the post-stimulus potentials decreases. This prediction is consistent with the recent observation that increasing the contrast of oriented bar stimuli reduces the variance in the postsynaptic response of orientation-specific neurons in macaque visual cortex [29]. Furthermore, our model predicts that the nature of stimuli during learning will affect the impact of sensory cues on electrophysiological quantities and behavior: more reliable priors will cause a smaller influence of sensory inputs, while increasing stimulus reliability, e.g., stimulus intensity, would achieve the opposite effect. Regardless of training, our model also predicts decreasing influence of the prior for increasing stimulus intensity.

## Gradient-based synaptic dynamics

As discussed above, a fixed stimulus determines the somatic membrane potential distribution. Prior to learning, this distribution will typically be different from a desired distribution as predicted, for example, by past sensory experience or cross-modal input. We refer to such stimulus-dependent desired distributions as target distributions.

We define learning in our framework as adapting synaptic weights $W$ to increase the probability of samples $u_s^*$ from the target distribution under the currently represented somatic posterior. Formally, learning reduces the Kullback-Leibler divergence $KL(p^*|p)$ between the target distribution $p^*(u_s|\boldsymbol{r})$ and the somatic membrane potential distribution $p(u_s|\boldsymbol{W}, \boldsymbol{r})$. This can be interpreted as a form of supervised learning, where a large divergence implies poor performance and a small divergence good performance, respectively. This is achieved through gradient ascent on the (log-)posterior somatic probability of target potentials $u_s^*$ sampled from the target distribution, resulting in the following dynamics for excitatory and inhibitory weights (for details see Methods Sec. "Weight dynamics"):

$$\dot{W}_i^{\mathrm{E/I}} \propto \lambda_e \frac{\partial}{\partial W_i^{\mathrm{E}}} \log p(u_s^*|\boldsymbol{W}, \boldsymbol{r}) \quad \propto \left[ \underbrace{(u_s^* - \bar{E}_s)(E^{\mathrm{E/I}} - \tilde{E}_i^{\mathrm{d}})}_{=\Delta\mu_i^{\mathrm{E/I}}} + \underbrace{\frac{\alpha_i^{\mathrm{sd}}}{2} \left( \frac{\lambda_e}{\bar{g}_s} - (u_s^* - \bar{E}_s)^2 \right)}_{=\Delta\sigma^2} \right] r_i , \quad (4)$$

with $\tilde{E}_i^{\mathrm{d}} = \alpha_i^{\mathrm{sd}} \bar{E}_s + (1 - \alpha_i^{\mathrm{sd}}) E_i^{\mathrm{d}}$. Here, $\lambda_e$ is the exploration parameter, $\alpha_i^{\mathrm{sd}}$ the an effective dendritic coupling strength, $E_i^{\mathrm{d}}$ the reversal potential of dendrite $i$ given by Eq 1, and $\bar{E}_s$ the total somatic reversal potential.

All dynamic quantities arising in the synaptic plasticity rule are neuron-local. The dendritic potentials $E_i^{\mathrm{d}}$ are available at the synaptic site, as well as the presynaptic rates $r_i$. We hypothesize that the backpropagating action potential rate that codes for $u_s^*$ can influence dendritic synapses [30]. Furthermore, the total conductance $\bar{g}_s$ determines the effective time constant by which the somatic membrane potential fluctuates and could be measured through its temporal correlation length. The exact molecular mechanisms by which these terms and their combinations are computed in the synapses remain a topic for future research.

## Joint learning of somatic mean and variance

The total postsynaptic error is composed of an error in the mean $\Delta\mu_i^{\mathrm{E/I}}$ and an error in the variance $\Delta\sigma^2$ (Eq 4). By jointly adapting the excitatory and inhibitory synapses, both errors in the

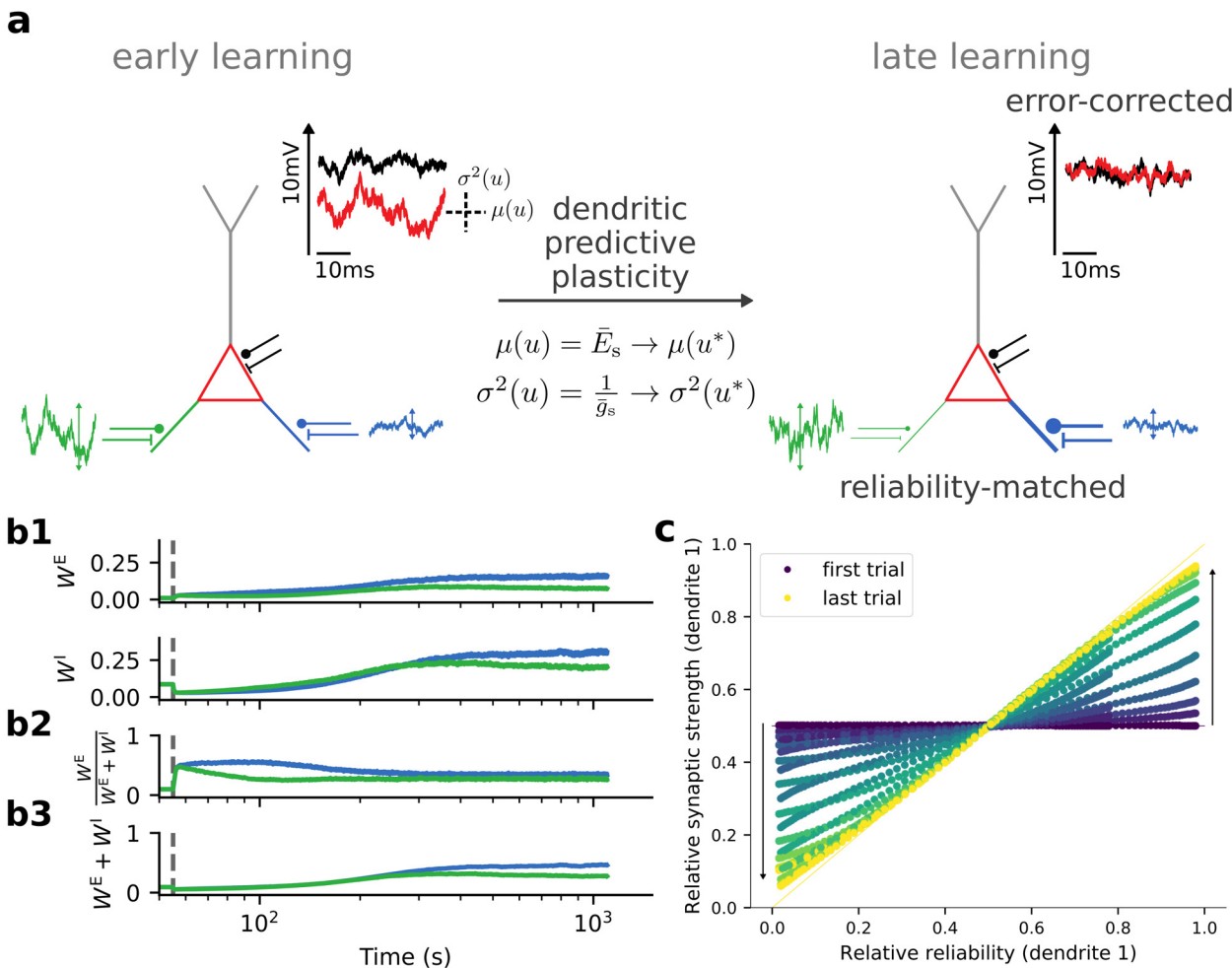

**Fig 6. Dendritic predictive plasticity performs error correction and reliability matching. (a)** A neuron receives input via two different input channels with different noise amplitudes (green and blue). Synaptic plasticity adapts the mean ($\mu$) and variance ($\sigma^2$) of the somatic membrane potential (red) towards the target (black). **(b1)** Excitatory and inhibitory weights per input channel (basal dendrite). The dashed vertical line indicates the onset of learning. The dendrites learn the mean target potential within the first few seconds (jumps after the dashed line). **(b2)** Ratio of excitatory and total synaptic weights per dendrite. These ratios determine the mean dendritic membrane potentials. Since both dendrites learn to match the same somatic mean potential based on their respective synaptic inputs, these ratios become equal. **(b3)** Sum of excitatory and inhibitory weights per dendrite. The total dendritic weights reflect the reliability of the dendritic input. Learning assigns larger synaptic weights to the less fluctuating and more reliable input (blue) as compared to the stronger fluctuating and less reliable input (green). As the balancing ratio becomes the same (b2), the excitatory and inhibitory strengths of the more reliable input must both become larger (b1). **(c)** The relative synaptic strength of a given branch ($W_i/\Sigma_j W_j$) becomes identical to the relative reliability ($\frac{1}{\sigma_i^2}/\sum_j \frac{1}{\sigma_j^2}$) of its input with respect to the other branches over the course of learning (here shown for $i = 1$; starting with $W_1 = W_2$ for the entire range of relative reliabilities, horizontal line). Note that time flows from blue (first trial) to yellow (last trial).

mean and the variance are reduced. To simultaneously adjust both the mean and variance, the two degrees of freedom offered by separate excitation and inhibition are required.

To illustrate these learning principles we consider a toy example in which a neuron receives input via two different input channels with different noise amplitudes. Initially neither the average somatic membrane potential, nor its variance match the the parameters of the target distribution (Fig 6a, left). Over the course of learning, the ratio of excitatory to inhibitory weights increases to allow the average somatic membrane potential to match the average target potential and the total strength of both excitatory and inhibitory inputs increases to match the

inverse of the total somatic conductance to the variance of the targets ([Fig 6a], right; b1). Excitatory and inhibitory weights hence first move into opposite directions to match the average, and later move in identical directions to match the variance ([Fig 6b1]).

In both dendrites, the strengths of excitation and inhibition converge to the same ratio to match the mean of the target distribution ([Fig 6b2]). However, the relative magnitude of the total synaptic strength $W^{\text{tot}} = W^{\text{E}} + W^{\text{I}}$ changes according to the relative fluctuations of the presynaptic input during learning. While branches with reliable presynaptic input (small fluctuations) are assigned large total synaptic weights, branches with unreliable input learn small total synaptic weights ([Fig 6b2]). More specifically, the total synaptic weights indeed match the respective reliabilities of the individual dendrites: $W^{\text{tot}} \propto \dfrac{1}{\sigma_r^2}$ ([Fig 6c]). Intuitively speaking, the total synaptic weights learn to modulate somatic background noise $\xi$ towards a target variance $\sigma_u^*$. For a proof, we refer to the SI.

## Learning Bayes-optimal cue combinations

We next consider a multisensory integration task in which a rat has to judge whether the angle of a grating is larger than 45˚ or not, using whisker touching (T) and visual inspection (V), see [Fig 7a] and [14]. In this example, projections are clustered according to modality on dendritic compartments. In general, this clustering is not necessarily determined by modality but could also reflect, for example, lower-level features, or specific intracortical pathways. In our setup, uncertainty in the sensory input from the two modalities is modeled by different levels of additive noise. The binary classification is performed by two multisensory output neurons that are trained to encode the features $> 45˚$ and $< 45˚$, respectively. Technically, we assume the target distribution is a narrow Gaussian centered around a stimulus-dependent target potential. For example, for the neuron encoding orientations $> 45˚$, the target potential would be high for ground truth orientations $> 45˚$ and it would be low otherwise. The output neurons receive input from populations of feature detectors encoding information about visual and tactile cues, respectively ([Fig 7b]).

The performance of the model neurons after learning matches well the Bayes-optimal MAP estimates that make use of knowledge about the exact relative noise variances. In contrast, averaging the two cues with equal weighting, and thus not exploiting the conductance-based Bayesian processing, or considering only one of the two cues, would result in lower

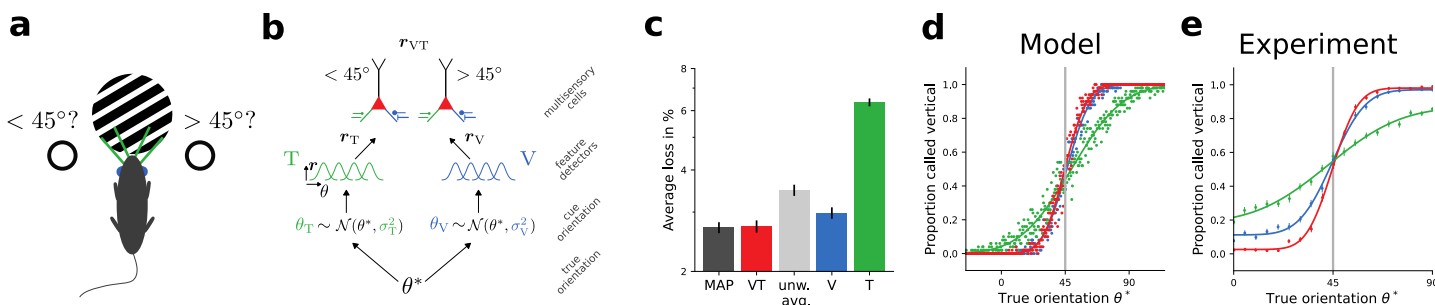

**Fig 7. Learning Bayes-optimal inference of orientations from multimodal stimuli. (a)** Experimental setup [14]. **(b)** Network model. **(c)** Accuracy of the MAP estimate (MAP, dark gray), the trained model with bimodal cues (VT, red), unweighted average of visual and tactile cues (unw. avg., light gray), and the trained model with only visual (V, blue) and tactile cues (T, green), respectively. Error bars denotes standard error of the mean over 25 experiments, each consisting of 20 000 trials. The trained model performs as well as a theoretically optimal observer (compare loss of MAP and VT). **(d)** Psychometric curves of the model confirm that the classification near 45˚ for the combined modalities (red) is at least as good as for the visual modality (V, blue, lower input variance), and better than for the tactile modality (T, green, higher input variability). Dots: subsampled data, solid lines: fit of complementary error function. **(e)** Psychometric curves for rat 1 [14] for comparison. Experiment data from [14].

performance (Fig 7c). Furthermore, the psychophysical curves of the trained model match well to experimental data obtained in a comparable setup (Fig 7d and 7e).

## Cross-modal suppression is caused by conductance-based Bayesian integration

Using the trained network from the previous section, we next consider the firing rate of the output neuron that prefers orientations $> 45°$ for conflicting cues with a specific mismatch. We assume a true stimulus orientation $> 45°$ generates a separate cue for each modality, where, as an example we assume the visual cue to be more vertical than the tactile cue (Fig 8a) which result in different dendritic reversal potentials $E_i^d$. In the following we identify the reliability of a stimulus with its intensity. Intuitively speaking, a weak stimulus is less reliable than a strong one.

When cues are presented simultaneously at low stimulus intensity, the output neurons fire stronger than in unimodal conditions (Fig 8b). However, when presented simultaneously at high stimulus intensity the cues suppress each other, i.e., the resulting firing rate is smaller than the maximal rate in unimodal conditions (Fig 8b). This phenomenon is known as cross-modal suppression [31, 32].

In the context of the conductance-based Bayesian integration, this counterintuitive interaction of multimodal cues arises as a consequence of the somatic potential being a weighted average of the two unimodal effective reversal potentials and the prior. For low stimulus intensity the prior dominates; since the evidence from either modality is only weak, information arriving from a second modality always constitutes additional evidence that the preferred stimulus is present. Thus, the somatic potential is pulled farther away from the prior in the bimodal condition as compared to the unimodal one. For high stimulus intensity the prior does not play a role and the somatic potential becomes a weighted average of the two modality-specific effective reversal potentials. As one cue is more aligned with the neuron's preferred feature than the other, the weighted average appears as a suppression (Fig 8).

We propose that the computational principle of conductance-based Bayesian integration also underlies other variants of cross-modal suppression (e.g., [7, 31–33]), and also explains

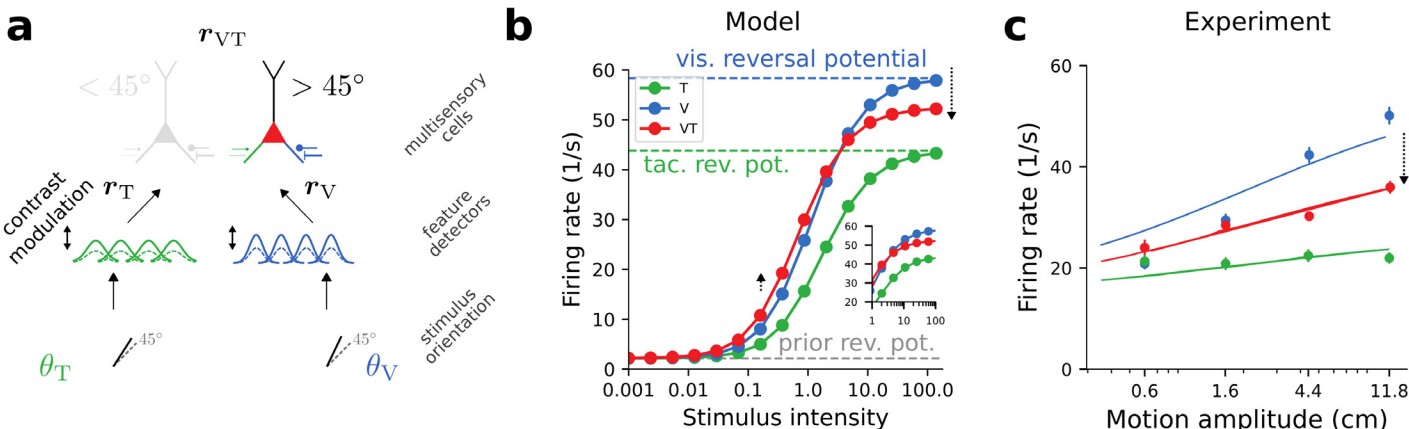

**Fig 8. Cross-modal suppression arising from Bayes-optimal integration of information in single neurons. (a)** Experimental setup (compare Fig 7). **(b)** Firing rate of the output neuron encoding orientations $> 45°$ for unimodal stimulation (V,T) and bimodal stimulation (VT). Dashed lines indicate the limit of no stimulation (gray), and infinitely strong tactile (green) and visual (blue) stimulation, respectively. Inset shows zoom in for high stimulation intensities. Pulling the somatic potential (red) towards the weighted mean of the visual and tactile effective reversal potentials (blue and green dashed lines) leads to a relative increase for weak stimulus intensities (black upward arrow) and to cross-modal suppression at strong stimulus intensities (black downward arrow). **(c)** Firing rate of a neuron from macaque MSTd in response to misaligned visual (blue) and vestibular (green) cues with a mismatch of $\Delta = 60°$. Experiment data from [31].

unimodal suppression arising from superimposing cues (e.g., [34–36]), or superimposing sensory inputs and optogenetic stimulation [37, 38].

## Discussion

The biophysics of cortical neurons can be interpreted as Bayesian computations. We demonstrated that the dynamics of conductance-based neuron models naturally computes posterior distributions from Gaussian likelihood functions and prior represented in dendritic compartments. We derived somatic membrane dynamics from stochastic gradient ascent on this posterior distribution, and synaptic plasticity from matching the posterior to a target distribution. Our plasticity rule naturally accommodates the relative reliabilities of different pathways by scaling up the relative weights of reliable inputs, i.e., those that have a high correlation to target potentials for given presynaptic activities. The targets may themselves be formed by perisomatic input from other modalities, or by more informed predictive input from other cortical areas. We demonstrated successful learning in a multisensory integration task in which modalities were different in their reliability.

Cortical and hippocampal pyramidal neurons have also been described to be driven by two classes of inputs, with general 'top-down' input on apical dendrites that predicts the 'bottom-up' input on basal dendrites [39, 40]. In this framework, adapting the basal inputs has been conceptualized as "learning by the dendritic prediction of somatic firing" [30, 41, 42]. In the broader context of our Bayesian framework, this view suggests that synaptic plasticity tries to match bottom up input to top-down expectations. Depending on the nature of the top-down input, learning can be thus interpreted as target matching or—in the absence of targets—as a regularization of the cortical representation similar to prior matching in variational autoencoders [43].

Our supervised learning can be seen within this predictive framework. A neuron is considered as a nonlinear prediction element, with dendritic input predicting somatic activity. Extending this predictive view, we argue that dendrites themselves can be seen as performing a dendritic 'opinion pooling' [24, 44], namely forming dendritic opinions on the stimulus feature, weighting them according to their reliability, and predicting the somatic opinion that is imposed by the teacher input. Each dendrite receives a subset of the neuron's afferents and forms its own opinion whether a certain feature is likely present in this afferent subset. While the dendritic opinion is encoded in the effective dendritic reversal potential, the reliability of this opinion is encoded in the total dendritic conductance. According to the biophysics of neurons, the overall somatic opinion is then formed by the certainty-weighted dendritic opinions, and this is what the somatic output represents.

So far, we have only considered synapses of which the conductance does not depend on the local membrane potential. Excitatory synapses in pyramidal cells are known to express N-methyl-D-aspartate (NMDA) channels, whose conductance depends on the local potential [45]. These synapses elicit strong supra-linear responses [46] which cause a massive increase of the isolated dendritic conductance and both dendritic and somatic potentials. In our current framework, such responses would correspond to a high certainty that a given feature is present in the input targeting the dendritic branch. Dendritic calcium spikes that originate in the apical dendrites of layer 5 pyramidal neurons [39, 47] may also represent such strong responses. At the time of the peak potential, when the derivative vanishes, these strong responses can be pooled with other dendritic potentials. As a result, the dendritic spikes can then be integrated according to their reliabilities to form the somatic posterior. However, these strongly non-linear, recurrent interactions are difficult to fully capture in the current mathematical framework. An extended model, which could also describe the influence of backpropagation action

potentials necessary for learning, is a promising direction to further reduce the gap to biophysical dynamics.

Bayesian inference has previously been suggested as an operation on the level of a neuronal population in space [3, 17, 48] or in time [12, 20, 21, 49]. In our framework, to read out the reliability of a single neuron, postsynaptic neurons either have to average across time or across a population of neurons that encode the same feature. Our single-neuron description of Bayesian inference may thus be complementary to population-based models. A formal demonstration of this complementarity is beyond the scope of the current manuscript. Other recent work also considers the neuronal representation and learning of uncertainty. For example, in line with our plasticity rules, natural-gradient-descent learning for spiking neurons [50] predicts small learning rates for unreliable afferents. A different approach to representing and learning uncertainty is centered on synaptic weights rather than membrane potentials and conductances [51]. In this model, each synapse represents a distribution over synaptic weights and plasticity adapts the parameters of this distribution. While being a complementary hypothesis, this normative view does not incorporate neuronal membrane dynamics.

Our model makes various experimental predictions.

(i) Certainty representation within a neuron: in response to individual whisker touches, our model implies that the somatic potential of somatosensory neurons is driven towards a stimulus-specific reversal potential; this is consistent with measurements in mouse barrel cortex (Fig 5). Moreover, the model also predicts that the variability of cumulative PSP amplitudes (jumps in the postsynaptic membrane potential following a whisker touch) depends on the frequency of whisker touches. For high frequencies, i.e., small inter-stimulus intervals, the total evoked conductance remains large and the somatic potential "sticks" more to the corresponding reversal potential between stimuli. Thus, the pre-stimulus variability of the somatic potential decreases, which in turn reduces the CV (coefficient of variation) of PSP amplitudes upon stimulation (consistent with experimental data, cf. Figs 1C & 6K in [25]). Similarly, we predict a drop in the CV of the PSPs with increased whisker deflection amplitude. A stronger, more certain stimulus would lead to stronger presynaptic firing; this consequently yields a stronger clamping and hence a smaller post-stimulus variability of the somatic potential, thereby reducing the variability of stimulus-induced PSPs.

(ii) Synaptic plasticity for certainty learning: to test whether the mean and variance of the somatic potential can be learned by dendritic input, one may consider extracellular stimulation of mixed excitatory and inhibitory presynaptic afferents of a neuron while clamping the somatic potential to a fluctuating target. Our plasticity rule predicts that initially, when the mean of the target distribution is not yet matched, excitatory and inhibitory synaptic strengths move in opposite directions, i.e., one increases, the other decreases, to jointly match the average somatic membrane potential to the target potential (cf. Fig 6b1). Then, after the match in the mean has been approximately reached, the excitatory and inhibitory strengths covary in order to match the variance of the target distribution.

(iii) Cross-modal suppression: consider a setting similar to [31] in which an animal receives mismatched visual and vestibular cues about a quantity of interest (cf. Fig 8). From a normative perspective, making the visual stimulus less reliable should shift weight to the vestibular input. Accordingly, our framework predicts that the total synaptic weights from the visual modality should become smaller. This causes visual cues to have a smaller effect on the somatic membrane potential, and thus, over the course of learning, the firing rate of the bimodal condition should become more similar to the tactile-only condition.

In conclusion, we suggest that single cortical neurons are naturally equipped with the 'cognitive capability' of Bayes-optimal integration of information. Moreover, our gradient-based formulation opens a promising avenue to explain the dynamics of hierarchically organized

networks of such neurons. Our framework demonstrates that the conductance-based nature of synaptic coupling may not be an artifact of the biological substrate, but rather enables single neurons to perform efficient probabilistic inference previously thought to be realized only at the circuit level.

## Methods

### Equivalent somato-dendritic circuit

The excitatory and inhibitory dendritic conductances, $g_i^{\mathrm{E}}$ and $g_i^{\mathrm{I}}$, are driven by the presynaptic firing rates $r_i(t)$ through synaptic weights $W_i^{\mathrm{E/I}}$ and have the form $g_i^{\mathrm{E/I}}(t) = W_i^{\mathrm{E/I}} r_i(t)$. For notational simplicity we drop the time argument in the following. The dynamics of the somatic potential $u_{\mathrm{s}}$ and dendritic potentials $u_i^{\mathrm{d}}$ for the $D$ dendrites projecting to the soma read as

$$C \dot{u}_{\mathrm{s}} = g_0(E_0 - u_{\mathrm{s}}) + \sum_{i=1}^{D} g_i^{\mathrm{sd}}(u_i^{\mathrm{d}} - u_{\mathrm{s}}) \tag{5}$$

$$C_i^{\mathrm{d}} \dot{u}_i^{\mathrm{d}} = g_i^{\mathrm{L}}(E^{\mathrm{L}} - u_i^{\mathrm{d}}) + g_i^{\mathrm{E}}(E^{\mathrm{E}} - u_i^{\mathrm{d}}) + g_i^{\mathrm{I}}(E^{\mathrm{I}} - u_i^{\mathrm{d}}) + g_i^{\mathrm{ds}}(u_{\mathrm{s}} - u_i^{\mathrm{d}}) \,, \tag{6}$$

where $C$ and $C_d$ are the somatic and dendritic capacitances, $E^{\mathrm{L/E/I}}$ the reversal potentials for the leak, the excitatory and inhibitory currents, $g_i^{\mathrm{sd}}$ the transfer conductance from the $i$th dendrite to the soma, and $g_i^{\mathrm{ds}}$ in the reverse direction. By $g_0$ and $E_0$ we denote the somatic conductance and its induced reversal potential, which in the absence of synaptic input to the soma becomes the leak conductance and the leak reversal potential.

We assume that $C^{\mathrm{d}}$s are small, so that dendritic dynamics are much faster than somatic dynamics and can be assumed to be in equilibrium. We can thus set $\dot{u}_i^{\mathrm{d}}$ to zero and rearrange Eq 6 to obtain

$$u_i^{\mathrm{d}} - u_{\mathrm{s}} = \frac{g_i^{\mathrm{d}}}{g_i^{\mathrm{d}} + g_i^{\mathrm{ds}}} \left(E_i^{\mathrm{d}} - u_{\mathrm{s}}\right) \,, \tag{7}$$

with dendritic reversal potentials $E_i^{\mathrm{d}}$ given by Eq 1 and $g_i^{\mathrm{d}} = g_i^{\mathrm{E}} + g_i^{\mathrm{I}} + g_i^{\mathrm{L}}$. Plugging Eqs 7 into 5 and using the shorthand notation $\alpha_i^{\mathrm{sd}} = \frac{g_i^{\mathrm{sd}}}{g_i^{\mathrm{ds}} + g_i^{\mathrm{d}}}$, we obtain

$$C\dot{u}_{\mathrm{s}} = g_0(E_0 - u_{\mathrm{s}}) + \sum_{i=1}^{D} \alpha_i^{\mathrm{sd}} g_i^{\mathrm{d}}(E_i^{\mathrm{d}} - u_{\mathrm{s}}) \,, \tag{8}$$

compare Eq 3 in the main manuscript. These dynamics are equivalent to gradient descent $(-\partial \mathcal{E}/\partial u_{\mathrm{s}})$ on the energy function

$$\mathcal{E}(u_{\mathrm{s}}) = \frac{g_0}{2}(E_0 - u_{\mathrm{s}})^2 + \sum_{i=1}^{D} \frac{\alpha_i^{\mathrm{sd}} g_i^{\mathrm{d}}}{2}(E_i^{\mathrm{d}} - u_{\mathrm{s}})^2 \,, \tag{9}$$

which also represents the log-posterior of the somatic potential distribution, as we discuss below.

### Bayesian theory of somatic potential dynamics

Above, we have outlined a bottom-up derivation of somatic dynamics from the biophysics of structured neurons. In the following, we consider a probabilistic view of single neuron computation and demonstrate that this top-down approach yields exactly the same somatic membrane potential dynamics.

The assumption of Gaussian likelihoods and priors reflects the fact that the summation of many independent synaptic inputs generally yields a normal distribution, according to the central limit theorem and in agreement with experimental data [18]. We thus consider a prior distribution over $u_s$ of the form

$$p(u_s|E_0, g_0) = \frac{1}{Z_0} e^{-\frac{g_0}{2\lambda_e}(E_0 - u_s)^2} \,, \tag{10}$$

with parameters $\lambda_e$, $g_0$, $E_0$ and normalization constant $Z_0$. Similarly, we define the dendritic likelihood for $u_s$ as

$$p(E_i^d|u_s, g_i^d) = \frac{1}{Z_i^d} e^{-\frac{\alpha_i^{sd} g_i^d}{2\lambda_e}(E_i^d - u_s)^2} \,, \tag{11}$$

with parameters $\alpha_i^{sd}, E_i^d, g_i^d$. According to Bayes' rule, the posterior distribution of the somatic membrane potential $u_s$ is proportional to the product of the dendritic likelihoods and the prior. If we further assume that dendrites are conditionally independent (independence of dendritic densities given the somatic potential), their joint density $p(\boldsymbol{E}^d|u_s, \boldsymbol{g}^d)$ factorizes, yielding

$$p(u_s \mid E_0, g_0, \boldsymbol{E}^d, \boldsymbol{g}^d) \propto p(\boldsymbol{E}^d \mid u_s, \boldsymbol{g}^d) p(u_s|E_0, g_0) = \prod_{i=1}^{D} p(E_i^d|u_s, g_i^d) p(u_s|E_0, g_0) \,. \tag{12}$$

Plugging in Eqs 10 and 11, we can derive that the posterior is a Gaussian density over $u_s$ with mean

$$\bar{E}_s = \frac{g_0 E_0 + \sum_{i=1}^{D} \alpha_i^{sd} g_i^d E_i^d}{g_0 + \sum_{i=1}^{D} \alpha_i^{sd} g_i^d} \tag{13}$$

and inverse variance

$$\bar{g}_s = g_0 + \sum_{i=1}^{D} \alpha_i^{sd} g_i^d \,. \tag{14}$$

We thus obtain

$$p(u_s|\boldsymbol{W}, \boldsymbol{r}) \equiv p(u_s \mid E_0, g_0, \boldsymbol{E}^d, \boldsymbol{g}^d) = \frac{1}{Z} e^{-\frac{\bar{g}_s}{2\lambda_e}(u_s - \bar{E}_s)^2} \,, \tag{15}$$

with normalization factor $Z = \sqrt{\frac{2\pi\lambda_e}{\bar{g}_s}}$. We switched in Eq 15 to the conditioning on $\boldsymbol{W}$ and the presynaptic rates $\boldsymbol{r}$ since these uniquely determine the dendritic and somatic conductances ($\boldsymbol{g}^d$), and thus also the corresponding reversal potentials ($\boldsymbol{E}^d$). Here, we use the conventional linear relationship $\boldsymbol{g} = \boldsymbol{W}\boldsymbol{r}$ between conductances and presynaptic rates. For more complex synapses with nonlinear transmission of the type $g = f(\boldsymbol{w}, \boldsymbol{r})$, where $f$ can be an arbitrary function, our derivation holds similarly, but would yield a modified plasticity rule.

The energy function from Eq 9 is equivalent to $\mathcal{E}(u_s) = -\lambda_e \log p(u_s|\boldsymbol{W}, \boldsymbol{r}) - \lambda_e \log Z = \frac{\bar{g}_s}{2}(u_s - \bar{E}_s)^2$. Since $Z$ is independent of $u_s$, the somatic membrane potential dynamics from Eq 8 minimizes the energy $\mathcal{E}$ while maximizing the log-posterior,

$$C\dot{u}_s = -\frac{\partial\mathcal{E}}{\partial u_s} = \lambda_e \frac{\partial}{\partial u_s} \log p(u_s|\boldsymbol{W}, \boldsymbol{r}) \,. \tag{16}$$

In this form, it becomes obvious that the somatic potential moves towards the maximum-a-posteriori estimate (MAP) of $u_s$ in the absence of noise. The stochastic version of Eq 16 with Gaussian additive noise leads to Eq 3 in the Results, this can be loosely interpreted as using Langevin dynamics to find the MAP solution for the posterior distribution.

## Weight dynamics

The KL between the target distribution $p^*$ and the somatic membrane potential distribution can be written as

$$KL[p^*(u_s|\boldsymbol{r})|p(u_s|\boldsymbol{W},\boldsymbol{r})] = -S(p^*) - \mathbb{E}_{p^*}[\log p(u_s|\boldsymbol{W},\boldsymbol{r})].\qquad(17)$$

The entropy $S$ of the target distribution $p^*$ is independent of the synaptic weights $\boldsymbol{W}$. Stochastic gradient descent on the KL divergence therefore leads to a learning rule for excitatory and inhibitory synapses that can be directly derived from Eq 15 (see SI):

$$\dot{W}_i^{\mathrm{E/I}} \propto \lambda_e \frac{\partial}{\partial W_i^{\mathrm{E/I}}} \log p(u_s^*|\boldsymbol{W},\boldsymbol{r}) = \alpha_i^{\mathrm{sd}}\left[\left(u_s^* - \bar{E}_s\right)\left(E^{\mathrm{E/I}} - \tilde{E}_i^{\mathrm{d}}\right) + \frac{\alpha_i^{\mathrm{ds}}}{2}\left(\frac{\lambda_e}{\bar{g}_s} - \left(u_s^* - \bar{E}_s\right)^2\right)\right] r_i,\quad(18)$$

with $\alpha_i^{\mathrm{sd}} = \frac{g_i^{\mathrm{sd}}}{g_i^{\mathrm{ds}}+g_i^{\mathrm{d}}}$, $\alpha_i^{\mathrm{ds}} = \frac{g_i^{\mathrm{ds}}}{g_i^{\mathrm{ds}}+g_i^{\mathrm{d}}}$ and $\tilde{E}_i^{\mathrm{d}} = \alpha_i^{\mathrm{ds}}\bar{E}_s + (1 - \alpha_i^{\mathrm{ds}})E_i^{\mathrm{d}}$, see also Eq 4 in the Results, where we assumed symmetric coupling conductances between dendritic compartments and soma, i.e., $g_i^{\mathrm{sd}} = g_i^{\mathrm{ds}}$.

As discussed in the main text, the two terms in the plasticity rule roughly correspond to adapting the mean and variance of the somatic distribution. However, the second term $\propto \frac{\lambda_e}{\bar{g}_s} - \left(u_s^* - \bar{E}_s\right)^2$ depends not only on a mismatch in the variance, but also on a mismatch in the mean of the distribution. To highlight this, we rewrite the sample $u_s^*$ as $u_s^* = \mu^* + \sigma^*\xi^*$, the target mean plus a sample from $\mathcal{N}(0,1)$ scaled with the target variance. Plugging this into the plasticity rule, the first term becomes $\propto (\mu^* + \sigma^*\xi^* - \bar{E}_s)$, and the second term becomes $\propto \frac{\lambda_e}{\bar{g}_s} - (\mu^* + \sigma^*\xi^* - \bar{E}_s)^2$. This form shows that only after the somatic reversal matches the target mean, $\bar{E}_s = \mu^*$, will the synapses adapt so that in expectation $\frac{\lambda_e}{\bar{g}_s} - (\sigma^*\xi^*)^2 \approx 0$. Because the $\xi^*$ are samples from a standard normal distribution, we conclude that after learning, beside $\bar{E}_s = \mu^*$, we also have $\frac{\lambda_e}{\bar{g}_s} = \sigma^{*2}$, i.e., the total synaptic conductance is inversely proportional to the variance of the target potential distribution. For a proof that, in addition, the total synaptic strength on each dendritic branch becomes inversely proportional to the variance in the presynaptic rate, $W^{\mathrm{tot}} \propto \frac{1}{\sigma_r^2}$, see SI.

In the absence of a target distribution, the neuron essentially sets its own targets. On average, weight changes in the absence of a target distribution are hence zero. Since for conductance-based synapses only non-negative weights are meaningful, we define the minimal synaptic weight as zero.

## Linear coordinates for nonlinear processing

The interplay of conductances and potentials can be visualized in a Cartesian plane spanned by inhibitory and excitatory conductances (Fig 9). To simplify the picture, we neglect leak conductances and assume strong dendritic couplings $g^{\mathrm{sd}}, g^{\mathrm{ds}}$. The state of a single dendrite is fully determined by its inhibitory and excitatory synaptic conductances and can be represented by a vector $(g^{\mathrm{I}}, g^{\mathrm{E}})$. As we assume the prior conductance is zero, the total conductance at the soma is given by the sum of dendritic conductances. Thus, the soma itself can be represented by a

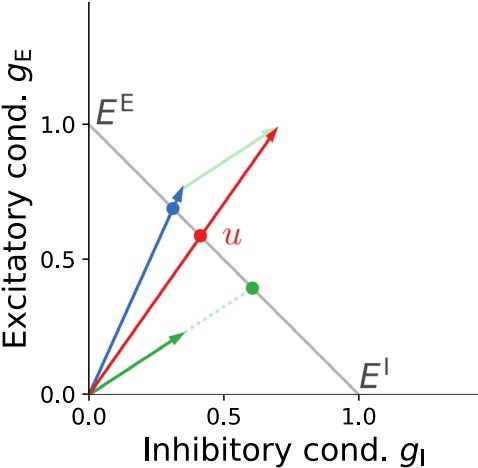

**Fig 9. The nonlinear membrane potential and synaptic dynamics expressed in linear conductance coordinates.** Dendrites can be represented as vectors defined by their inhibitory and excitatory conductances (blue and green arrows). In these coordinates, the soma is itself represented by a vector that is simply the sum of dendritic vectors (red arrow). The antidiagonal (gray) spans the range of all possible membrane potentials, from $E^I$ to $E^E$. The membrane potential of any given compartment is given by the intersection of its conductance vector with the antidiagonal.

vector that is the sum of the dendritic conductance vectors. Furthermore, the length of these vectors is proportional to the magnitude of excitatory and inhibitory conductances and thus the reliability of the potential encoded by their associated compartments.

This simple, linear construction also allows us to determine the membrane potentials of individual compartments. For this, we need to construct the antidiagonal segment connecting the points $(1, 0)$ and $(0, 1)$. If one identifies the endpoints of this segment with the synaptic reversal potentials, i.e., $E^I \to (1, 0)$ and $E^E \to (0, 1)$, the antidiagonal can be viewed as a linear map of all possible membrane potentials. With this construction, the membrane potential of a compartment (dendritic or somatic) is simply given by the intersection of its conductance vector with the antidiagonal. Formally, this intersection is a nonlinear operation and instantiates a convex combination, the core computation that connects neuronal biophysics to Bayesian inference (Fig 3).

This simple construction allows us to easily visualize the effects of synaptic weight changes on the dendritic and somatic membrane potentials. For example, increasing the inhibitory conductance of a certain compartment will have a twofold effect: its effective reversal potential will decrease (the intersection will move towards $E^I$), while simultaneously increasing its reliability (the vector will become longer).

In the following, we give a simple geometric proof that the intersection $u$ of a conductance vector $(g^I, g^E)$ with the antidiagonal indeed represents the correct membrane potential of the compartment. The coordinates of this intersection are easy to calculate as the solution to the system of equations that define the two lines $x/y = g^I/g^E$ and $y = 1 - x$, with

$$(x, y) = \left( \frac{g^I}{g^I + g^E}, \frac{g^E}{g^I + g^E} \right).$$ (19)

The ratio of these coordinates is also the ratio of the two resulting segments on the

antidiagonal: $(E^{\mathrm{E}} - u)/(u - E^{\mathrm{I}}) = x/y$. Solving for $u$ yields

$$u = \frac{g^{\mathrm{I}} E^{\mathrm{I}} + g^{\mathrm{E}} E^{\mathrm{E}}}{g^{\mathrm{I}} + g^{\mathrm{E}}} \,, \tag{20}$$

which represents the sought convex combination.

## Simulation details

In the following we provide additional detail on simulations. Numerical values for all parameters can be found in Tables 1–4.

**Details to Fig 5.** We consider the trained network from Fig 7, but now use a finite somatic capacitance $C$. The differential equation of the output neurons (Eq 3) is integrated on a time grid of spacing $\Delta t$ with an explicit Runge-Kutta method of order 3(2) from SciPy 1.4.1 [52]. To mimic background noise we generate "noise" cues, identical for both modalities, from a normal distribution $\mathcal{N}(\mu_{\mathrm{b}}, \sigma_{\mathrm{b}}^2)$ and convert these into rates $r^{\mathrm{b}}$ via the two populations of feature detectors. We consider an additional "signal" cue, also identical across modalities and trials, which generates additional rates $r'$ via the feature detectors. The input rate for the output neurons is then computed as $r = \gamma r' + (1 - \gamma)r^{\mathrm{b}}$, where $\gamma = \gamma^{\mathrm{before}}$ before stimulus onset and $\gamma = \gamma^{\mathrm{after}}$ after stimulus onset. For visualization purposes, we shift the scale of membrane potentials by $-8$mV in the figure.

**Details to Fig 6.** We consider a neuron following instantaneous versions of Eq 3. It has $D$ compartments with infinitely strong coupling of the dendritic compartments to the soma $g^{\mathrm{ds}}$, $g^{\mathrm{sd}} \to \infty$. In each trial, we sample a ground truth input rate $r \sim \mathcal{N}(\mu_r, \sigma_r^2)$, and from this rate we generate noisy rates $r^{\mathrm{V}} \sim \mathcal{N}(r, \sigma_{\mathrm{V}}^2), r^{\mathrm{T}} \sim \mathcal{N}(r, \sigma_{\mathrm{T}}^2)$ with modality-specific noise amplitudes $\sigma_{\mathrm{V}}, \sigma_{\mathrm{T}}$, respectively. We avoid non-positive input rates by replacing them with $r_{\mathrm{min}}$. We introduce an additional neuron with just a single compartments which generates target membrane potentials $u^*$ from the ground truth input rate $r$ and a random weight matrix. The second neuron receives the noisy input rates and should learn to mimic the distribution of somatic target potentials by learning synaptic weights via Eq 4. We train for a certain number of trials $N_{\mathrm{trials}}$, and for visualization purposes convert trial number into time by defining a trial duration of $\Delta t_{\mathrm{trial}}$.

**Details to Fig 7.** We consider $N$ output neurons each with $D$ dendritic compartments. Their dynamics are described by Eq 3, but for computational efficiency we consider an instantaneous version of with $C \to 0$. We furthermore assume infinitely strong coupling of the dendritic compartments to the soma $g^{\mathrm{ds}}, g^{\mathrm{sd}} \to \infty$. We use a softplus activation function $\rho(u_{\mathrm{s}}) = \log(1 + \exp(u_{\mathrm{s}}))$.

**Table 1. Parameters used in Fig 5.** Remaining parameters defined in Table 3.

| Parameter name | Value | Description |
|---|---|---|
| $N_{\mathrm{trials}}$ | 40 | number of trials |
| $\mu^{\mathrm{noise}}, \sigma^{\mathrm{noise}}$ | 35˚, 15˚ | mean/std. of noise orientations |
| $\theta_{\mathrm{stimulus}}$ | 44˚ | stimulus orientation |
| $\gamma^{\mathrm{before}}, \gamma^{\mathrm{after}}$ | 0.0, 0.88 | rel. signal contrast before/after stimulus onset |
| $dt$ | 0.2 ms | integration time step |
| $T$ | 100 ms | simulation duration |
| $C$ | 50 pF | somatic membrane capacitance |
| $\lambda_{\mathrm{e}}$ | 100.0 nS mV$^2$ | neuronal exploration constant |

**Table 2. Parameters used in Fig 6.** Remaining parameters defined in Table 3.

| Parameter name | Value | Description |
|---|---|---|
| $N$ | 1 | number of neurons |
| $D$ | 2 | number of dendritic compartments per neuron |
| $g_0^{\mathrm{L}}$ | 0.25 nS | somatic leak conductance |
| $g_i^{\mathrm{L}}$ | 0.025 nS | dendritic leak conductance |
| $w_{\mathrm{init}}^{\min}, w_{\mathrm{init}}^{\max}$ | 0.0 nS s, 0.019 nS s | min/max value of initial excitatory weights |
| $w_{\mathrm{init}}^{\min}, w_{\mathrm{init}}^{\max}$ | 0.0 nS s, 0.21 nS s | min/max value of initial inhibitory weights |
| $w_{\mathrm{init}}^{\min}, w_{\mathrm{init}}^{\max}$ | 0.0 nS s, 1.07 nS s | min/max value of target excitatory weights |
| $w_{\mathrm{init}}^{\min}, w_{\mathrm{init}}^{\max}$ | 0.0 nS s, 7.0 nS s | min/max value of target inhibitory weights |
| $\eta$ | $1.25 \cdot 10^{-3}$ | learning rate |
| $N_{\mathrm{trials}}$ | 110 000 | number of trials |
| $\Delta t_{\mathrm{trial}}$ | 10 ms | trial duration |
| $r^*$ | $\mathcal{N}\left(1.2\,\frac{1}{\mathrm{s}}, 0.5\,\frac{1}{\mathrm{s}}\right)$ | distribution of input rates |
| $r_{\min}$ | $0.001\,\frac{1}{\mathrm{s}}$ | minimal input rate |
| $\sigma_{\mathrm{T}}$ | $0.3\,\frac{1}{\mathrm{s}}$ | noise amplitude of tactile modality |
| $\sigma_{\mathrm{V}}$ | $0.01875\,\frac{1}{\mathrm{s}}$ | noise amplitude of visual modality |

**Table 3. Parameters used in Fig 7.**

| Parameter name | Value | Description |
|---|---|---|
| $N$ | 2 | number of neurons |
| $D$ | 3 | number of dendritic compartments per neuron |
| $g_0^{\mathrm{L}}$ | 1.0 nS | somatic leak conductance |
| $g_i^{\mathrm{L}}$ | 0.2 nS | dendritic leak conductance |
| $E^{\mathrm{E}}, E^{\mathrm{I}}$ | 0 mV, −85 mV | exc. /inh. reversal potentials |
| $E^{\mathrm{L}}$ | −70 mV | leak potential |
| $\lambda_{\mathrm{e}}$ | 1.0 nS mV$^2$ | neuronal exploration constant |
| $C$ | $\to 0$ | somatic membrane capacitance |
| $g_i^{\mathrm{sd}}, g_i^{\mathrm{ds}}$ | $\to \infty$ | somato-dendritic/dendro-somatic coupling conductance |
| $N_{\mathrm{T}}, N_{\mathrm{V}}$ | 70 | number of feature detectors per modality |
| $[\theta_{\min}^{\mathrm{fd}}, \theta_{\max}^{\mathrm{fd}}]$ | [−315˚, 405˚] | min/max preferred orientations of feature detectors |
| $\kappa$ | $6.0\,\frac{1}{\deg^2}$ | concentration (inverse variance) of feature detectors |
| $r_{\mathrm{low}}, r_{\mathrm{high}}$ | $0.75\,\frac{1}{\mathrm{s}}, 16.0\,\frac{1}{\mathrm{s}}$ | min/max rates of feature detectors |
| $w_{\mathrm{init}}^{\min}, w_{\mathrm{init}}^{\max}$ | 0.0 nS s, 0.005 nS s | min/max value of initial excitatory weights |
| $w_{\mathrm{init}}^{\min}, w_{\mathrm{init}}^{\max}$ | 0.0 nS s, 0.024 nS s | min/max value of initial inhibitory weights |
| $\eta$ | $0.25 \cdot 10^{-4}$ | learning rate |
| $\sigma_{\mathrm{T}}$ | 28.5˚ | tactile noise amplitude |
| $\sigma_{\mathrm{V}}$ | 13.5˚ | visual noise amplitude |
| $[\theta_{\min}^{\mathrm{train}}, \theta_{\max}^{\mathrm{train}}]$ | [−270˚, 360˚] | min/max of training orientations |
| $[\theta_{\min}^{\mathrm{test}}, \theta_{\max}^{\mathrm{test}}]$ | [−135˚, 225˚] | min/max of testing orientations |
| $\theta_{\mathrm{db}}$ | 45˚ | decision boundary |
| $N_{\mathrm{train}}$ | 400 000 | number of training trials |
| $N_{\mathrm{test}}$ | 500 000 | number of testing trials |
| $p_{\mathrm{bimodal}}$ | 0.9 | probability of a bimodal trial during training |
| $b$ | 12 | batch size |
| $r_{\mathrm{low}}^*, r_{\mathrm{high}}^*$ | $0.75\,\frac{1}{\mathrm{s}}, 16.0\,\frac{1}{\mathrm{s}}$ | low/high target rates |

**Table 4. Parameters used in Fig 8.** Remaining parameters defined in Table 3.

| Parameter name | Value | Description |
|---|---|---|
| $\theta_T$ | 65˚ | orientation of tactile cue |
| $\theta_V$ | 50˚ | orientation of visual cue |
| $c_T, c_V$ | $[10^{-3}, 10^2]$ | stimulus contrasts of tactile and visual modality |
| $r_{scale}$ | 2.5 | output rate scaling factor |

We define two homogeneous input populations of $N_T$ and $N_V$ feature detectors, respectively, with Gaussian tuning curves. The output rate of a feature detector in response to a cue with orientation $\theta$ is given by:

$$r(\theta) = r_{min} + (r_{max} - r_{min})e^{-\frac{\kappa}{2}(\theta - \theta')^2} , \qquad (21)$$

with minimal rate $r_{min}$, maximal rate $r_{max}$, concentration $\kappa$ and preferred orientation $\theta'$. The preferred orientations $\theta'$ are homogeneously covering the interval $[\theta_{min}^{fd}, \theta_{max}^{fd}]$. All feature detectors from one population project to one dendritic compartment of each output neuron via plastic connections.

Each output neuron additionally receives an input from one presynaptic neuron with fixed rate but plastic weight, allowing it to adjust its prior expectations.

Initial weights are randomly sampled from a zero-mean normal distribution with standard deviation $\sigma_{init}^w$. Training proceeds as follows. From a ground-truth orientation $\theta^*$ two cues, $\theta_V$, and $\theta_T$, are generated by sampling from a Gaussian distribution around a true stimulus value with modality-specific noise amplitudes $\sigma_V$ and $\sigma_T$). The true orientation $\theta^*$ determines the output neurons target rates and hence, via the inverse activation function, target membrane potentials. The output neuron which should prefer orientations $> 45$˚ is trained to respond with a rate $r_{low}^*$ if $\theta < 45$˚ and with a rate $r_{high}^*$ if $\theta \geq 45$˚. The other output neuron is trained in the opposite fashion. Weight changes are following Eq 4. To speed up training we use batches of size $b$ for $N_{train}$ trials with ground truth orientations $\theta^*$ sampled uniformly from $[\theta_{min}^{train}, \theta_{max}^{train}]$. During training, with probability $p_{bimodal}$ cues are provided via both modalities, while $1 - p_{bimodal}$ of all trials are unimodal, i.e., feature detectors of one modality remain silent.

For testing the output neurons are asked to classify $N_{test}$ cues uniformly sampled from $[\theta_{min}^{test}, \theta_{max}^{test}]$, again perturbed by modality specific noise. The classification is performed on the combined rate of the two output neurons $r = 0.5(r_0 + (r_{low} + r_{high} - r_1))$, where $r_0$ is the rate of the neuron preferring orientations $> 45$˚ and $r_1$ the rate of the other output neuron. A ground truth orientation $\theta^*$ is classified as $>= 45$˚ if $r >= r_{low} + 0.5(r_{high} - r_{low})$.

**Details to Fig 8.** We consider the trained network from Fig 7. Here we set the cues provided to the feature detectors of the tactile and visual modality to fixed values $\theta_V$, $\theta_T$, respectively. We introduce two additional parameters, the stimulus intensities $c_V$, $c_T$, which linearly scale the rates of all feature detectors of the respective modality. For visualization purposes we scale the rate of the output neuron by a factor $r_{scale}$.

## Supporting information

**S1 Text.** 1. Definitions. 2. Derivation of the somatic potential distribution. 3. Derivation of membrane potential dynamics. 4. Derivation of weight dynamics. 5. Unreliable dendritic inputs are assigned small synaptic strengths. 6. Dendritic parameters.
(PDF)

## Acknowledgments

WS thanks M. Larkum and F. Helmchen for many inspiring discussions on dendritic processing, and M. Diamond and N. Nikbakht for sharing and discussing their data in an early state of this work. The authors thank all members of the CompNeuro and NeuroTMA groups for valuable discussions.

## Author Contributions

**Conceptualization:** Jakob Jordan, João Sacramento, Mihai A. Petrovici, Walter Senn.

**Data curation:** Jakob Jordan, Willem A. M. Wybo.

**Formal analysis:** Jakob Jordan, João Sacramento, Willem A. M. Wybo, Mihai A. Petrovici, Walter Senn.

**Funding acquisition:** Mihai A. Petrovici, Walter Senn.

**Investigation:** Jakob Jordan, Willem A. M. Wybo.

**Methodology:** Jakob Jordan, João Sacramento, Willem A. M. Wybo, Mihai A. Petrovici, Walter Senn.

**Project administration:** Mihai A. Petrovici, Walter Senn.

**Resources:** Mihai A. Petrovici, Walter Senn.

**Software:** Jakob Jordan, Willem A. M. Wybo.

**Supervision:** Mihai A. Petrovici, Walter Senn.

**Validation:** Jakob Jordan, Mihai A. Petrovici.

**Visualization:** Jakob Jordan, Willem A. M. Wybo.

**Writing – original draft:** Jakob Jordan, João Sacramento, Mihai A. Petrovici, Walter Senn.

**Writing – review & editing:** Jakob Jordan, João Sacramento, Willem A. M. Wybo, Mihai A. Petrovici, Walter Senn.

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
