## [Decision Letter · Decision Letter 0]

31 Mar 2023

Dear Dr. Jordan,

Thank you very much for submitting your manuscript "Learning Bayes-optimal dendritic opinion pooling" for consideration at PLOS Computational Biology.

As with all papers reviewed by the journal, your manuscript was reviewed by members of the editorial board and by several independent reviewers. In light of the reviews (below this email), we would like to invite the resubmission of a significantly-revised version that takes into account the reviewers' comments.

The authors should attempt to address all of the reviewer comments. Most importantly, they should try to greatly improve the clarity of the manuscript. They could consider asking colleagues unfamiliar with the work to read it and identify any areas that are hard to follow.

We cannot make any decision about publication until we have seen the revised manuscript and your response to the reviewers' comments. Your revised manuscript is also likely to be sent to reviewers for further evaluation.

Sincerely,

Blake A Richards

Academic Editor

PLOS Computational Biology

Thomas Serre

Section Editor

PLOS Computational Biology

The authors should attempt to address all of the reviewer comments. Most importantly, they should try to greatly improve the clarity of the manuscript. They could consider asking colleagues unfamiliar with the work to read it and identify any areas that are hard to follow.

Reviewer's Responses to Questions

**Comments to the Authors:**

Reviewer #1: Review uploaded as an attachment

Reviewer #2: I believe I understand Bayesian inference pretty well, especially in the Gaussian setting, but I did not understand what's going on in this paper until I sat down and figured it out on my own. And even then I'm guessing I don't have the full story. Following is what I _think_ the authors did.

In the end, it's kind of a simple story. Sort of. Let's start with a review of Bayesian inference when all distributions are Gaussian. If we have a set of observations, E_i, with variance sigma^2_i, and the prior is Normal(x_0, sigma_0^2), then, defining

g_i = 1/sigma_i^2,

the posterior is Gaussian with:

mean = (sum_i g_i E_i) / (sum_i g_i)

1/variance = sum_i g_i

where the sum starts at i=0 (to take into account the prior).

So if g_i is conductance and E_i is reversal potential, the mean looks a lot like the mean voltage in a conductance based model (so long as the neuron doesn't spike).

But reversal potentials mainly come in two flavors: around 0 mV for excitatory input and around -80 mV for inhibitory input. So how do we get an in-between membrane potential? From a weighted sum of excitatory and inhibitory inputs. The relative weights give the mean reversal potential; to get the overall strength you have to match the conductance (via the above equation).

All this has to be learned, of course, and the authors write down a learning rule (Eq. 5). It's reasonable, but they don't derive it. I looked in Methods, but got a bit lost because I couldn't tell the difference between a scalar, vector and matrix, so I gave up. Without going through the details, though, it's hard to tell if this makes sense, especially since the firing rate distribution is going to be constantly changing. And to learn they need samples from the true posterior -- it's totally unclear where that would come from. (The authors say backpropagating action potentials, but that doesn't really help, because then the soma has to know the posterior.)

What's really needed is a cost function one can wrap one's head around, like performance on a task, rather then the KL divergence. With the right cost function and problem setup, everything would be easy to derive.

On top of that, the scheme requires the dendritic time constant to be much faster than the somatic one. Which it isn't; they're about the same. Does this kill the idea?

So my take is: it's possible that the authors have done something sensible, but their explanation was so inscrutable to me that I couldn't evaluate it. In my youth I probably would have gone through Methods and SI and figured it out, but that's really not my job. If the other reviewers are happy, I'm certainly OK with this being published, although my predictions is that very few people will ever understand what they did. However, it would be nice if they wrote the paper clearly enough that it could be easily understood by your run-of-the mill theorist with a decent math background. I don't think it is now. (Or maybe I just need to be a lot smarter -- don't want to rule that out.)

Reviewer #3: The current study investigates the Bayesian computation in a single conductance-based neuron, where the membrane potential represents the posterior mean, and the conductance represents the posterior precision. In this way of representation, the single neuron dynamics can implement the Bayesian computation well. The authors also considered some tasks to demonstrate the performance of the model. Overall, I feel this work provide new insight onto Bayesian computation on single neurons, and I like the geometric interpretation of the Bayesian computation (Fig. 3).

Major:

- The current model assumes that each dendritic compartment has an associated preferred feature (text below Eq. 1). Considering a single only has finite number of dendritic compartments, I am wondering how a neuron could represent and conduct the computation of the distribution over continuous variables? Does this require an idealized neuron with infinite number of dendritic compartments? If I misunderstood, how the digital to analog conversion can be done at single neuron level? Some discussions would be helpful.

- For simulations in Fig. 7: the text mentions that the uncertainty in the sensory input is modeled by different levels of additive noise. On the other hand, the current framework the likelihood uncertainty is represented by conductance in a dendritic compartment which is proportional to input firing rate, which means the sensory input firing rate encodes the input uncertainty. Therefore, I am puzzled without altering the input firing rate (with the same conductance) but just changing the input additive noise, how the single neuron could do the Bayesian computation optimally?

- Deneve, Neural Computation 2008a, b also studied the Bayesian computation in single neurons which is quite relevant to the current study. It might be good to compare the current work with Deneve’s.

Minor:

- The text above Eq. 15: is it should be the inverse of variance?

- The text after Eq. 17: the Langevin sampling from the posterior seems to be over claimed. Implementing the Langevin sampling is not merely injecting noise into the gradient ascent dynamics on the posterior, but it has a stronger requirement that the drift and fluctuations terms share the same factor. The current manuscript doesn’t provide any support about this.

**Have the authors made all data and (if applicable) computational code underlying the findings in their manuscript fully available?**

Reviewer #1: Yes

Reviewer #2: Yes

Reviewer #3: Yes

PLOS authors have the option to publish the peer review history of their article (what does this mean?). If published, this will include your full peer review and any attached files.

Reviewer #1: No

Reviewer #2: No

Reviewer #3: **Yes: **Wen-Hao Zhang
---

## [Decision Letter · Decision Letter 1]

2 Jan 2024

Dear Dr. Jordan,

Thank you very much for submitting your manuscript "Conductance-based dendrites perform Bayes-optimal cue integration" for consideration at PLOS Computational Biology.

As with all papers reviewed by the journal, your manuscript was reviewed by members of the editorial board and by several independent reviewers. In light of the reviews (below this email), we would like to invite the resubmission of a significantly-revised version that takes into account the reviewers' comments.

The revision represents an improvement but Reviewer 2 remains unconvinced and, in fact, is frustrated by the difficulty of understanding fundamental points. Please carry out all the requested remaining revisions and, in writing a reply, please describe in detail how you have responded, including quotes from your revisions (as opposed to simply saying you have revised, and having us examine the revision). This will make it easier for us to see exactly what has changed concerning each item and how you are understanding the need to revise. Note that Reviewer 2 continues to feel the necessary revisions are "major." That is a judgment call, but we do need to be convinced that you have made a good faith effort to address the points raised, and that you have been largely successful in doing so. Thank you for your attention to these matters as they seem to go beyond good exposition, which is of course also important on its own.

We cannot make any decision about publication until we have seen the revised manuscript and your response to the reviewers' comments. Your revised manuscript is also likely to be sent to reviewers for further evaluation.

[1] A letter containing a detailed list of your responses to the review comments and a description of the changes you have made in the manuscript, as described above. Please note while forming your response, if your article is accepted, you may have the opportunity to make the peer review history publicly available. The record will include editor decision letters (with reviews) and your responses to reviewer comments. If eligible, we will contact you to opt in or out.

Sincerely,

Robert E. Kass

Guest Editor

PLOS Computational Biology

Thomas Serre

Section Editor

PLOS Computational Biology

Reviewer's Responses to Questions

**Comments to the Authors:**

Reviewer #1: Thank you very much for responding to my comments and suggestions. I found all responses acceptable and that the revised version of the manuscript addresses my previous concerns. The clarity of the manuscript has improved.

(5), (6), (11) are sufficiently addressed in the discussion.

(7), (8), (9), (12) are sufficiently addressed in the results.

The language in question for (10) and (14) has been removed.

I agree with the authors for (13) that this is not an issue.

The claims, scope and limitations suggested and highlighted are effectively addressed in (14) and (15).

Reviewer #2: In my previous review, my main complaint was that the paper was very hard to make sense of. That hasn't changed a lot; the paper is still very hard to make sense of. This time, however, I took a closer look at Methods (the _only_ way to figure out what was actually going on). I think I sort of know what they're doing, but now things don't exactly make sense to me.

So here's a quick summary: information about the true membrane potential of a neuron come from "dendrites" (by which they mean dendritic branches?). Because they're using conductance-based synapses, each dendrite encodes the mean and variance of a Gaussian distribution. That mean and variance is combined optimally by the post-synaptic neuron.

So here's what I find problematic:

1. After digging through supplementary materials, I found that for the excitatory and inhibitory conductances, the authors use

g_i^E = W_i^E r

g_i^I = W_i^I r

where r is the (global?) firing rate. At least I think that's what they use; I had to infer that from Eq. 13 of supplementary materials (I couldn't find it explicitly stated anywhere else, but maybe I missed it?). But this isn't what happens on real neurons; instead, the membrane potential on a dendrite is driven by many different inputs, from many different neurons. So it should be

g_i^E = sum_j W_{ij}^E r_j^E

g_i^I = sum_j W_{ij}^I r_j^I

This means the conductances -- and thus the mean and variance of the likelihood on dendrite i -- is determined not just by the weights, but also by the firing rates of different populations of neurons. That's fine, but it needs a _much_ more sophisticated learning rule than the one given in the paper. This is a central issue that needs to be addressed.

2. Given the setup, I was expecting a learning rule that would ensure that each dendritic branch carried the correct mean and variance. But that's not what the learning rule, Eq. 4, seems to do. For instance, take the case in which u*_s is fixed. In that case, the learning rule would first drive the difference in the excitatory and inhibitory weights so that \\bar{E}_s = u*_s, then drive magnitude of both to infinity (to make \\bar{g}_s as large as possible). Moreover, the weights would be infinity independent of the reliability of the different dendrites. Which seems odd to me.

At least that's what the learning rule says. But on page 9 the authors say:

"To illustrate these learning principles we consider a toy example in which a neuron receives input via two different input channels with different noise amplitudes. Initially neither the average somatic membrane potential, nor its variance match the the parameters of the target distribution (Fig. 6a, left). Over the course of learning, the ratio of excitatory to inhibitory weights increases to allow the average somatic membrane potential to match the average target potential and the total strength of both excitatory and inhibitory inputs increases to match the inverse of the total somatic conductance to the variance of the targets (Fig. 6a, right; b1)."

This implies that they can indeed learn the correct mean and variance. But I simply could not figure out how.

And, as an aside, the authors say "We hypothesize that the backpropagating action potential rate that codes for u*_s can influence dendritic synapses [30]." But the backpropagating action potential rate is just the firing rate of the neuron. How would the neuron know how to set its firing rate to match u*_s?

3. The authors assume that different dendrites are independent. However, we now know that that's a bad idea, and, at least in some circumstances, correlations totally dominate when it comes to information (Information-limiting correlations, Moreno-Bote, et al., Nature Neurosci. 17:1410-1417 2014). So this seems like a seriously problematic assumption.

4. Finally, it's not clear why the brain cares about p(u*_s). Doesn't it want a probability distribution over task-relevant variables? I'm not saying it _doesn't_ want p(u*_s), I'm just saying it's not obvious to me why it does. The authors need to motivate this a little better.

The bottom line is slightly worse than last time: I now have several reservations about the paper. However, I strongly suspect that I didn't fully understand it (which in itself is a problem), so possibly the reservations are easily addressed.

Reviewer #3: I am overall satisfied about the revisions made by the authors, while I suggest the authors make below revisions.

- Please refrain from using the Langevin sampling below Eq. 16 because the current method only outputs the MAP rather than the whole posterior distribution. An alternative might be "... using Langevin dynamics to find the MAP solution...". In addition, the authors may be interested in a nonlinear circuit implementation of Langevin sampling https://doi.org/10.1101/2020.07.20.212126

- I admit that the representation in current study is quite distinct with Deneve 2008. But it would be quite beneficial for readers to compare different studies if the author could discuss this briefly in Discussion.

**Have the authors made all data and (if applicable) computational code underlying the findings in their manuscript fully available?**

Reviewer #1: Yes

Reviewer #2: Yes

Reviewer #3: None

PLOS authors have the option to publish the peer review history of their article (what does this mean?). If published, this will include your full peer review and any attached files.

Reviewer #1: **Yes: **Ilenna Simone Jones

Reviewer #2: No

Reviewer #3: **Yes: **Wenhao Zhang
---

## [Editor Report · Decision Letter 2]

31 Mar 2024

Dear Dr. Jordan,

We are pleased to inform you that your manuscript 'Conductance-based dendrites perform Bayes-optimal cue integration' has been provisionally accepted for publication in PLOS Computational Biology. The guest editor apologizes for his confusion at the previous iteration.

Best regards,

Robert E. Kass

Guest Editor

PLOS Computational Biology

Thomas Serre

Section Editor

PLOS Computational Biology

---

## [Editor Report · Acceptance letter]

26 Apr 2024

PCOMPBIOL-D-22-01785R2 

Conductance-based dendrites perform Bayes-optimal cue integration

Dear Dr Jordan,

I am pleased to inform you that your manuscript has been formally accepted for publication in PLOS Computational Biology. Your manuscript is now with our production department and you will be notified of the publication date in due course.

With kind regards,

Anita Estes
